# *Blastocystis* spp. Infection in Kidney Transplant Recipient

**DOI:** 10.3390/pathogens14040341

**Published:** 2025-04-01

**Authors:** Justyna Kaczmarek, Małgorzata Marchelek-Myśliwiec, Danuta Kosik-Bogacka, Joanna Korycińska, Małgorzata Lepczyńska, Grażyna Dutkiewicz, Joanna Kabat-Koperska

**Affiliations:** 1Department of Nephrology, Transplantology and Internal Medicine, Pomeranian Medical University, Powstańców Wielkopolskich 72, 70-111 Szczecin, Poland; malgorzata.marchelek.mysliwiec@pum.edu.pl (M.M.-M.); grazyna.dutkiewicz@pum.edu.pl (G.D.); askodom@poczta.onet.pl (J.K.-K.); 2Department of Biology, Parasitology and Pharmaceutical Botany, Pomeranian Medical University, Powstańców Wielkopolskich 72, 70-111 Szczecin, Poland; danuta.kosik.bogacka@pum.edu.pl; 3Department of Medical Biology, School of Public Health, Collegium Medicum, University of Warmia and Mazury in Olsztyn (UWM), Żołnierska 14C, 10-561 Olsztyn, Poland; joanna.korycinska@uwm.edu.pl (J.K.); mlepczynska@gmail.com (M.L.)

**Keywords:** *Blastocystis* spp., kidney transplantation, diarrhea, infection

## Abstract

The *Blastocystis* sp. is a common enteric parasite found in humans and various animals. *Blastocystis* spp. infections may be asymptomatic or symptomatic, with gastrointestinal and extra-intestinal symptoms, such as diarrhea, nausea, abdominal pain, bloating, vomiting, or anorexia. The disease leading to symptoms is usually observed in participants with immune deficiency. We report the case of weight loss and diarrhea in a *Blastocystis* sp. infection in a 64-year-old renal transplant recipient.

## 1. Introduction

Kidney transplantation (KTx) is the treatment of choice in end-stage kidney disease as it prolongs the life of patients and improves quality of life in comparison to dialysis sessions [1,2,3]. The introduction of new immunosuppressive drugs has significantly extended survival time in individuals after KTx, but we observe many side effects of immunosuppressive therapy in our daily routines during regular check-up visits with patients in a Transplantology Outpatient Clinic. In kidney transplant patients, like in other solid organ transplant (SOT) recipients, one of the most common symptoms is diarrhea, which may result from the side effects of immunosuppressive medications or gastrointestinal disorders of infectious or non-infectious etiologies. The standard immunosuppressive regimen in SOT recipients includes medications from the group of calcineurin inhibitors (most commonly tacrolimus), mycophenolate derivatives, and glucocorticosteroids. Both tacrolimus and mycophenolate mofetil (or mycophenolate sodium) can cause diarrhea in the post-transplant period. In such cases, the treatment involves reducing the doses of the aforementioned medications [4]. Other non-infectious causes of diarrhea in organ transplant recipients may include non-specific inflammatory bowel diseases, such as ulcerative colitis and Crohn’s disease, mechanical complications of surgery, and metabolic conditions [5,6].

In cases of a suspected infectious etiology of diarrhea, diagnostic testing for the most common gastrointestinal pathogens, including bacterial, fungal, viral, and parasitic infections, is recommended [7,8]. In kidney transplant recipients, infections caused by protozoa, including *Entamoeba histolytica*, *Giardia duodenalis*, *Cyclospora* spp., *Cystoisospora* spp., *Enterocytozoon bieneusi*, and *Cryptosporidium* spp., as well as helminths (*Taenia solium*, *Ascaris lumbricoides*, *Strongyloides stercoralis*, *Trichuris trichiura*, and *Dipylidium caninum*) have been described [9,10]. Intestinal protozoan infections with *Blastocystis* spp., whose pathogenesis is not fully understood, are also frequently observed in these patients [11,12]. Parasites, such as *Blastocystis* spp. and *Cryptosporidium* spp., are often asymptomatic or responsible for infections with mild symptoms in healthy populations but may cause prolonged and heavy infections with gastrointestinal complaints, mainly diarrhea, in immunocompromised patients. Such infections can often not be detected by routine diagnostic procedures, but special concentration and staining methods are needed [13]. Organ transplant recipients may acquire parasitic infections in three different ways, such as a transmission with the graft, de novo infection, or the reactivation of a latent infection as a consequence of immunosuppression [14]. The aim of this study is to discuss a case of a kidney transplant recipient infected with *Blastocystis* spp.

## 2. Case Presentation

A 64-year-old male with a diagnosed chronic kidney disease secondary to hypertensive nephropathy, who underwent kidney transplantation four years ago, was referred to the hospital due to persistent abdominal pain lasting approximately two months, significant weight loss, and diarrhea. Laboratory tests revealed stable creatinine levels (1.4 mg/dL and eGFR 53 mL/min/1.73 m^2^) and elevated inflammatory markers (CRP 25.28 mg/L, leukocytes 16.43 G/L, neutrophils 11.09 G/L, and monocytes 1.58 G/L). On physical examination, a midline abdominal hernia and tenderness in the midepigastric region were noted (the patient had a previous unsuccessful attempt at hernia reduction). Moreover, the patient’s medical history included post-transplant diabetes mellitus (PTDM), nodular gastropathy, and a previous *Helicobacter pylori* infection (status post eradication therapy). The treatment regimen consisted of prednisone (1 × 5 mg), tacrolimus (2 mg in the morning and 1 mg in the evening), mycophenolate mofetil (2 × 500 mg), and glimepiride (1 × 2 mg).

During hospitalization, ultrasound imaging revealed an anechoic lesion in the pancreas; no abnormalities were noted in the liver or pancreatic parameter levels. An abdominal X-ray ruled out gastrointestinal perforation and obstruction. A contrast-enhanced abdominal CT showed pancreatic changes described as being consistent with cysts. Further observation revealed an increase in inflammatory markers: CRP 207.22 mg/L, PCT 1.54 ng/mL, leukocytes 31.31 G/L, neutrophils 25.21 G/L, and monocytes 2.89 G/L. Blood and urine cultures were collected, and empirical antibiotic therapy with ceftriaxone and metronidazole was initiated.

The blood and urine cultures were negative. Infections with rotaviruses, adenoviruses, *Clostridium difficile*, CMV, and BKV were excluded. Following the implemented treatment, a gradual reduction in inflammatory markers was observed (CRP 33.91 mg/L, leukocytes 11.85 G/L, neutrophils 6.62 G/L, and monocytes 1.68 G/L), along with an improvement in the general condition and a reduction in pain symptoms. The patient was discharged home with a recommendation to return for a scheduled hospitalization one month later for follow-up tests and was further advised to continue metronidazole for 5 days, supplemented with a probiotic.

The patient returned for the follow-up in accordance with the recommendations within the prescribed time. Laboratory tests showed a renewed increase in inflammatory markers (CRP 88.75 mg/L, leukocytes 24.92 G/L, neutrophils 18.98 G/L, and monocytes 2.19 G/L). Elevated renal parameters were also noted (creatinine 2.04 mg/dL and eGFR 33 mL/min/1.73 m^2^), along with reduced iron and folic acid levels and significantly increased calprotectin (>800 µg/g). Gastroscopy revealed grade I esophageal varices, a hiatal hernia, and gastric mucosal inflammation with erosions. A colonoscopy showed areas of reddened mucosa with a blurred vascular pattern and small scattered erosions in the sigmoid colon. A histopathological examination of the gastric biopsy revealed chronic inflammation of the mucosa without signs of metaplasia. In the colon, the findings showed oedematous mucosa with ulcers, disrupted glandular architecture, an inflammatory infiltrate with a predominance of neutrophils, and the presence of occasional crypt abscesses. *Salmonella* spp., *Shigella* spp., *Campylobacter* spp., and *Yersinia* spp. infections and an infection of a fungal etiology were not confirmed. A microscopic stool specimen examination revealed the presence of numerous developmental forms of *Blastocystis* spp. (Figure 1). The conventional polymerase chain reaction (cPCR) analysis of the stool sample identified *Blastocystis* subtype 3 (ST3). The nucleotide sequence (PV366952) obtained in this study were deposited in GenBank.

Sulfasalazine (2 × 1000 mg) and tinidazole (4 × 500 mg for 2 days) were introduced to the treatment regimen. A significant clinical improvement was observed, with a complete resolution of symptoms. Follow-up laboratory tests demonstrated a marked reduction in inflammatory markers (CRP 36.69 mg/L, leukocytes 9.92 G/L, neutrophils 4.74 G/L, and monocytes 1.11 G/L) and improvement in renal function parameters (creatinine 1.16 mg/dL and eGFR 66 mL/min/1.73 m^2^). Stool microscopy was performed three times one month after treatment. The stool examination revealed no presence of developmental forms of *Blastocystis* spp. Inflammatory markers remained low.

In Table 1 below, the laboratory findings from both mentioned hospitalizations are shown.

During a planned check-up visit in the Outpatient Transplantology Ward one year later the patient remained symptomless and inflammatory markers were low but residual, in comparison to previous tests—there were significantly less single developmental forms of *Blastocystis* spp. found in the microscopic stool specimen examination. 

## 3. Discussion

*Blastocystis* spp. are anaerobic, unicellular, and intestinal parasitic protists able to colonize the large intestine of many vertebrate species, including humans [15,16]. It presents multiple evolutionary stages of life cycles, i.e., vacuolar, granular, multi-vacuolar, a-vacuolar, ameboid, and cystic forms. In the stool of the described kidney transplant recipient, single vacuolar forms were detected, which are the most commonly observed forms in the stool of infected individuals [17]. *Blastocystis* spp. exhibit a high degree of genetic variability. Based on the molecular analysis of the small subunit ribosomal RNA (SSU rRNA) gene, at least 42 subtypes (STs) of *Blastocystis* spp. have been identified from various animals and humans. In humans, 16 subtypes (ST1–ST10, ST12, ST14, ST16, ST23, ST35, and ST41) have been reported [18]. Determining the subtype is important in epidemiological studies. ST1–ST4 comprise 91.65% of all STs identified in humans globally [19]. The prevalence of Blastocystis spp. differs between geographical regions, countries, and communities and may reach from 2.5% to 56% in developed countries and 70 or 80% in developing countries. In Poland its prevalence was found to range from 0.14% to 23.6% [20].

*Blastocystis* sp. has been described as an opportunistic pathogen and has recently been implicated as an important cause of diarrheal illness in immunocompromised individuals [21,22]. In renal transplant recipients it has been found that *Blastocystis* spp. infection occurs in 3% to 39.1% cases [13,14,23,24]. But not only patients treated with immunosuppressive drugs after kidney transplantation are prone to this opportunistic infection. It has been found that hemodialysis patients, due to a dysfunction of the immune response in this end-stage renal disease, were infected with *Blastocystis* spp. at a rate of 13.6% [25,26]. Because of the patient’s medical history, we know he is a member of a high-risk group, due to immunosuppressive treatments.

*Blastocystis* sp. infection most commonly occurs due to poor hygiene practices, contact with infected animals or patients, and the consumption of contaminated food or water [27]. Due to data from the interview, the patient has not had contact with wild animals or infected pets and his hygiene practices have been good.

*Blastocystis* sp. infection may be asymptomatic or symptomatic, with gastrointestinal and extraintestinal symptoms, such as diarrhea, nausea, abdominal pain, bloating, vomiting, or anorexia [12]. It has been noted that *Blastocystis* spp. infections may exacerbate irritable bowel syndrome and inflammatory bowel diseases [28,29]. *Blastocystis* spp. have the ability to regulate the gut microbiome [30]. This effect is suspected to result not from a direct pro-inflammatory action but from the pathogen’s influence on the immune system or the composition of the gut microbiome, which plays a significant role in the development of inflammatory bowel diseases [31].

The laboratory diagnosis for *Blastocystis* spp. is based on a direct microscopic examination of fecal material, with or without the addition of Lugol’s iodine solution, and a permanent smear stained with trichrome. Xenic in vitro cultures are effective due to their higher sensitivity and specificity, but this is time consuming method. A polymerase chain reaction (PCR) is an effective technique for the diagnosis of a *Blastocystis* infection; its limitations are that it is expensive, time consuming, and labor-intensive because of the manual extraction of the DNA and the need for specialized equipment [32]. Establishing an unambiguous diagnosis in that case requires a variety of microbiological tests, from those available we excluded infections caused by rotaviruses, adenoviruses, *Clostridium difficile*, CMV, BKV, *Salmonella* spp., *Shigella* spp., *Campylobacter* spp., *Yersinia* spp. and an infection of a fungal etiology. The microscopic stool specimen examination revealed the presence of numerous developmental forms of *Blastocystis* spp. We confirmed the ST3 subtype, one of the most common according to many sources and probably the most frequent subtype of *Blastocystis hominis* [19,33,34]. It is still debated if the *Blastocystis* subtype has a real impact on the clinical manifestation. There are reports that mention that the subtype can cause the symptomatic course of a *Blastocystis* infection, has impact on the gut microbiota, and in some cases is connected with IBS (irritable bowel syndrome) and IBD (inflammatory bowel disease) [35,36,37].

According to previous information, the described case concerns the development of a symptomatic *Blastocystis* spp. infection in a patient with a weakened immune system due to immunosuppressive therapy. During the first hospitalization, stool samples for parasitological testing were not collected; however, metronidazole was administered, which contributed to suppressing the infection, even though it was undiagnosed. During the second hospitalization, the *Blastocystis* sp. was correctly diagnosed, and different treatment was initiated. Following the causal treatment (metronidazole), a significant clinical improvement was not observed—shortly after treatment symptoms were reported again; test results confirmed higher rates of inflammatory parameters and the presence of the featured pathogen. A permanent recovery, defined by lack of symptoms and normal values of inflammatory parameters, was obtained by tinidazole. In the microscopic stool specimen examination we found single, but residual developmental forms of *Blastocystis* spp., which did not cause a symptomatic infection. The resistance to typical treatments is reported in the literature and a connection with geographical location or different subtypes of *Blastocystis* spp. is suggested. In some cases, a change in medication or combination of several drugs is supposedly needed to achieve improvement [38,39,40,41]. Nevertheless, it is difficult to definitively determine whether the patient’s symptoms were solely related to the *Blastocystis* spp. infection or if they were induced by the infection triggering an underlying inflammatory bowel disease within the gastrointestinal tract or if they were induced by the wrong drug choice or a treatment that was too short.

It is worth noting that a symptomatic *Blastocystis* spp. invasion may also occur after transplantation, when immunosuppressive drug doses have already been reduced, and such cases are also described in the literature [9]. Routine examinations of stool samples for parasites would significantly benefit the renal transplant recipients by contributing to the reduction of severe infections, which was underlined in former studies [12,22].

In conclusion, the role of the *Blastocystis* sp. as a pathogenic factor responsible for gastrointestinal and extra-intestinal symptoms may be underestimated in clinical practice. Infections with this protozoan are often overlooked in diagnostics. Immunocompromised patients, including organ transplant recipients undergoing immunosuppressive therapy, represent a high-risk group for *Blastocystis* spp. infection.

According to the literature sources, we know that the role of *Blastocystis hominis* is not ambiguous, and although the course of the disease, symptoms, and test results in the described patient are consistent with the characteristics of a *Blastocystis hominis* infection, we cannot be certain that the confirmed infection was the only cause of the symptomatic infection. It is clear that this case needs further investigation among immunocompromised patients.

## Figures and Tables

**Figure 1 pathogens-14-00341-f001:**
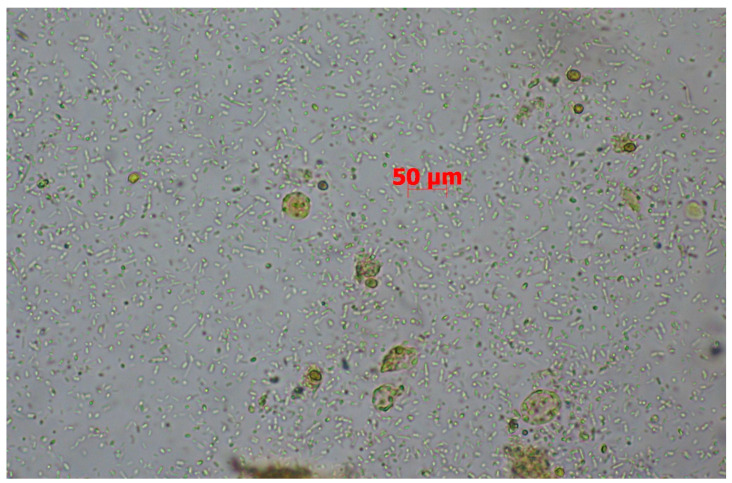
The vacuolar forms of *Blastocystis* spp. in fecal materials (40× magnification).

**Table 1 pathogens-14-00341-t001:** The laboratory test results. The hospitalization number is added in parentheses.

Parameter	Laboratory Normal Range	Day						
		1 (1)	2 (1)	3 (1)	6 (1)	7 (1)	1 (2)	4 (2)
Leukocytes (G/μL)	4.00–10.00	16.43	29.38	31.31	9.82	11.85	24.92	9.92
Neutrophils (G/μL)	2.00–6.90	11.09	23.20	25.21	5.10	6.62	18.98	4.74
Monocytes (G/μL)	0.10–0.90	1.58	2.93	2.89	1.44	1.68	2.19	1.11
CRP (mg/L)	<5.0	25.28	94.40	207.22	47.61	33.91	88.75	36.69
Procalcitonin (ng/mL)	<0.05		0.37	1.54				
Creatinine (mg/dL)	0.70–1.20	1.40		1.71	1.03		2.04	1.16
GFR (mL/min/m^2^)	80–120	53.0		41.0	76.0		33.0	66.0
Tacrolimus (ng/mL)	3.00–8.00	5.80			4.30			5.60

## Data Availability

The data presented in this study are available on request from the corresponding author due to privacy restrictions.

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
