# Peer review of "Blastocystis spp. Infection in Kidney Transplant Recipient"

_pathogens, 2025, doi:10.3390/pathogens14040341_

Round 1
Reviewer 1 Report
Comments and Suggestions for Authors
Dear Authors,
This case report presents interesting findings. Blastocystis infection was eradicated, but the patient was likely reinfected. The resolution of symptoms following antibiotic treatment may be attributed to changes in the bacterial composition of the colon. Please discuss this point in relation to articles on Blastocystis treatment. I think that the discussion section is the weakest part of the article. Additionally, subtype determination was not performed, which could be considered a limitation.
- Revise the phrase "limited infections in normal."
- Please add a scale to the image.
- Clarify the statement: "The laboratory findings from both hospitalizations are shown in Table 1." Emphasize the aspect of both hospitalizations.
- In the discussion section, there is irrelevant information about Blastocystis and the case. Consider summarizing the first three paragraphs into a single paragraph.
Sincerely
Author Response
Thank you very much for your suggestions. We have taken them into account to improve our work.

Reviewer 2 Report
Comments and Suggestions for Authors
The case report entitled " Blastocystis spp. Infection in a Kidney Transplant Recipient" represents a considerable amount of work. The following comments need to be addressed.
- Blastocystis should be italic throughout the manuscript.
- Line 117: I propose adding this reference: PMID: 38003823
- If possible, it is essential to conduct molecular characterization and sequencing analysis of this isolate to determine the subtype associated with this case.
Author Response

(The authors gave the same response as above.)

Reviewer 3 Report
Comments and Suggestions for Authors
The authors present a case study of a kidney transplant patient in which a Blastocystis infection was detected. Treatment of this infection alleviated the clinical manifestations. Blastocystis is an extremely common infection of the gastrointestinal tract that is generally asymptomatic with mild symptoms. The immunosuppression associated with the transplantation may be the cause of the clinical manifestations, and the authors provided references about other examples of immunosuppression causing symtoms associated with Blastocystis.
Overall, the paper is well written and accurate. Due to the commonness of Blastocystis, it may be a concern among transplant patients and merits publication. My comments are very minor:
line 72, 00
line 111, Ward (not Word)
line 173, delete later
Author Response
Thank you very much for yours suggestions. With no doubts we will take them into account to improve our work.
